# Low-Profile Broadband Filtering Antennas for Vehicle-to-Vehicle Applications

**DOI:** 10.3390/s25154747

**Published:** 2025-08-01

**Authors:** Shengtao Chen, Wang Ren

**Affiliations:** School of Information and Electronic Engineering, Zhejiang Gongshang University, Hangzhou 310018, China; 22020090045@pop.zjgsu.edu.cn

**Keywords:** filtering antenna, broadband, Vehicle-to-Vehicle, characteristic mode analysis

## Abstract

This paper proposes a compact, broadband, and low-profile filtering antenna designed for Sub-6 GHz communication. By applying characteristic mode analysis to the radiating elements, the operational mechanism of the antenna is clearly elucidated. The current cancellation among different radiating elements results in two radiation nulls in the primary radiation direction, effectively enhancing the filtering effect. The antenna achieves a wide operational bandwidth ($S_{11}\leq -10$ dB) of 35.9% (4.3–6.4 GHz), making it highly suitable for Sub-6 GHz communication systems. Despite its compact size of 25 × 25 mm^2^, the antenna consistently maintains stable broadside radiation patterns, with a peak gain of 6.14 dBi and a minimal gain fluctuation of less than 1 dBi at 4.6–6.45 GHz. This design ensures reliable and robust communication performance for V2V systems operating in the designated frequency band.

## 1. Introduction

The rapid development of the electric vehicle industry continues to elevate vehicle intelligence, expanding in-vehicle systems to encompass applications like autonomous driving, intelligent transportation, and fleet management. The core components of intelligent driving, Vehicle-to-Everything (V2X) and Vehicle-to-Vehicle (V2V) [1,2,3,4] communications, demand communication systems that provide low latency and high bandwidth to enable timely safety features such as emergency braking warnings, collision avoidance, and traffic flow control.

The 5.9 GHz and 4.9 GHz frequency bands play a crucial role in V2X and V2V communications. The design of a filter antenna that covers both frequency bands significantly enhances anti-jamming capabilities, ensuring efficient communication between vehicles and infrastructure. This not only improves the performance of autonomous driving and intelligent transportation systems but also guarantees the precise transmission of real-time data in complex environments. This paper focuses on designing a Sub-6 GHz filter antenna that accommodates the 5.9 GHz and 4.9 GHz frequency bands to fulfill the communication demands of V2X and V2V. The antenna will optimize anti-jamming performance and bandwidth, ensuring efficient data transmission, which ultimately improves the overall performance of autonomous driving and intelligent transportation systems.

To achieve a Sub-6 GHz filter antenna suitable for V2V communication, several requirements need to be met, including miniaturization and broadband performance. Integrating filtering and antenna functions together significantly reduces the system size and weight compared with traditional separate designs. Common methods for implementing filter antennas include dielectric resonator patch slotting [5,6,7,8], but these generally result in narrow bandwidth, high design precision requirements, and strict production processes. Short-circuit via loading [9] also suffers from the issue of narrow bandwidth. Stacked patches [10,11,12,13,14,15] are another common approach, but the miniaturization results are not as effective. This paper adopts the method of adding connected patch antennas, which can simultaneously increase bandwidth and introduce null points, meeting both miniaturization and filtering requirements.

To design a broadband filtering antenna for the Sub-6 GHz band, Characteristic Mode Analysis (CMA) can be used as an analytical tool. CMA helps designers identify the different characteristic modes of the antenna, enabling effective control over the antenna’s radiation characteristics and frequency response. In the Sub-6 GHz design, CMA allows for a systematic analysis of the antenna’s operating modes, identifies potential filtering zeros, and helps achieve broadband and filtering functionality by adjusting the geometry of the patches and adding parasitic patches.

This approach allows designers to optimize antenna performance more precisely, extending the operating bandwidth and achieving the desired filtering zeros at high and low frequencies without introducing complex structures. In this way, CMA not only enhances design efficiency but also ensures optimal antenna performance within the Sub-6 GHz frequency range.

## 2. Antenna Design

### 2.1. Theory of Characteristic Modes

Characteristic Mode Analysis (CMA) is a powerful method for investigating the fundamental electromagnetic behavior of conducting structures, especially in antenna design. It enables the decomposition of surface currents into a set of orthogonal characteristic modes, each associated with a distinct current distribution and resonant frequency, offering valuable insight into the radiation mechanism of the antenna.

The total surface current J(r) on a conducting body can be expressed as a linear combination of characteristic mode currents:(1)$$J\left(r\right)=\sum_{n}a_{n}J_{n}\left(r\right)\;$$
where $J_{n}\left(r\right)$ is the *n*-th normalized characteristic mode current, and $a_{n}$ is the excitation coefficient determined by the external feed.

The characteristic modes $J_{n}$ are obtained by solving the following generalized eigenvalue equation:(2)$$XJ_{n}=\lambda_{n}RJ_{n}$$

Here, Z=R+jX is the impedance operator from the Method of Moments (MoM), with R and X being its real and imaginary components, respectively. The eigenvalue $\lambda_{n}$ represents the resonance behavior of the *n*-th mode: the mode is resonant when $\lambda_{n}=0$.

To quantify how efficiently each mode radiates at a specific frequency, the Modal Significance (MS) is defined as:(3)$${\mathrm{MS}}_{n}=\left|\frac{1}{1+j\lambda_{n}}\right|$$

The value of ${\mathrm{MS}}_{n}$ ranges from 0 to 1, with ${\mathrm{MS}}_{n}=1$ indicating maximum radiation efficiency.

In this work, CMA is performed using the MoM-based solver in CST Microwave Studio. The antenna structure is simulated with an infinitely extended ground plane and dielectric substrate in the *x*-*y* plane. By evaluating modal currents, eigenvalues, and MS curves, we identify dominant radiation modes and guide the structural optimization of the antenna for broadband and filtering performance.

### 2.2. Design Evolution

From Figure 1 for antenna I, at the resonance point of the MS curve, approximately around a frequency of f = 6.2 GHz, the current distribution reveals that the mode consists predominantly of a simple array composed of two half-wavelength dipoles. Given that the current directions of these two dipoles are coherent, as shown in Figure 2, their far-field radiation reinforces each other, thus increasing the overall radiation intensity. Although the antenna demonstrates a relatively broad bandwidth at this juncture (indicated by MS > 0.7), its coverage in the low-frequency spectrum (<5 GHz) is inadequate. Consequently, to improve performance in the low-frequency band, it is essential to further refine the antenna structure to extend the operating frequency range, with particular emphasis on improving low-frequency coverage.

Based on Antenna I, Antenna II integrates parasitic patches on both sides. Simulation results from the MS curve show that Antenna II operates in two modes: a low-frequency Mode 1 at around 4.4 GHz and a high-frequency Mode 2 at about 6.2 GHz, indicating bandwidth potential. In Mode 1, the current is mainly on the central patch, but the parasitic patches extend the half-wavelength current path, lowering the resonance frequency. From an equivalent circuit view, these patches increase the capacitance of the system, further reducing the resonance frequency. In Mode 2, the current concentrates on the side patches, maintaining a half-wavelength resonance. Note that the current direction of the central patch and the parasitic patches is opposite, potentially creating radiation nulls to enhance the antenna’s selectivity.

Considering the single-layer design with coaxial feeding in the center patch, parasitic patches positioned far from the center lead to weak coupling. To enhance this, a central connecting line was added to bridge the central patch with the side parasitic patches, resulting in Ant III. The simulation results for Ant III align closely with Ant II, maintaining two modes: Mode 1 and Mode 2. In Mode 2, current primarily flows through the side parasitic patches, maintaining a half-wavelength resonance. In Mode 1, the central connecting line creates a folded half-wavelength resonance with four segments. The observed current cancellation in opposite directions suggests the potential for radiation nulls in the low-frequency band.

### 2.3. Mechanism of Radiation Nulls

Radiation nulls are formed when destructive interference occurs between different parts of the radiating structure, typically due to oppositely directed surface currents. These nulls are essential in achieving out-of-band radiation suppression and enhancing frequency selectivity.

As illustrated in Figure 3a, at around 4.2 GHz (Mode 1), the surface currents on the central patch and side parasitic patches flow in opposite directions. This leads to cancellation of far-field radiation in the broadside direction, resulting in a low-frequency radiation null. Specifically, the current path forms a folded half-wavelength resonance, with four segments contributing to current cancellation.

As the frequency increases, the current amplitude on the central patch grows stronger. At approximately 6.8 GHz (Mode 2), as shown in Figure 3b, the side parasitic patches carry currents that are nearly equal in magnitude but opposite in phase compared with those on the central patch. This again causes destructive interference in the broadside direction, forming a second radiation null in the high-frequency band.

These radiation nulls are not incidental but are closely tied to the excitation and interference of specific characteristic modes. Their presence is key to the antenna’s filtering functionality, enabling strong attenuation at out-of-band frequencies while preserving desired in-band performance. The use of CMA allows for precise identification and control of such modal behaviors during the design phase.

## 3. Antenna Geometry

The geometric structure of the antenna designed for the Sub-6 GHz frequency band is shown in Figure 4. This antenna consists of two metal layers: the antenna radiation surface on top and the ground plane below. The entire structure is made of an FR-4 dielectric substrate with a thickness of 3 mm. It comprises four patches and three stepped impedance resonators.

In the center of the structure, there are two identical rounded rectangular structures. The central rectangular section has a length of RL1 and a width of RW1, with two semicircles of diameter RW1 added to the top and bottom. The side patches are rounded rectangles formed by a rectangular section with a length of RL2 and a width of RW2, along with two semicircles of diameter RW2. The patches are connected by three-stepped impedance resonators, with the middle resonator having dimensions SL1 and SW1, while the two adjacent resonators have identical dimensions of SL2 and SW2. The filtering antenna is excited by a 50 Ω SMA connector. The geometric dimensions are summarized in Table 1. Unless otherwise specified, these parameters will be used throughout the design.

## 4. Discussions on Simulated and Experimental Results

The antenna introduced in the previous section has been fabricated and measured to validate the proposed concept. The photographs of the fabricated prototype, including the assembly and soldering with coaxial cables, are shown in Figure 5a,b. Figure 6a and Figure 6b show the actual testing environments for $S_{11}$ and radiation gain measurements, respectively.

### 4.1. Return Loss and Gain

The simulated and measured S-parameters of the designed antenna are shown in Figure 7, which presents the simulated and measured broadside realized gain and reflection coefficient of the antenna. According to the simulation results, the low-frequency null occurs at 4.2 GHz, and the high-frequency null occurs at 6.8 GHz. However, in the actual measurements, two radiation nulls are observed at 4 GHz and 6.75 GHz. The measured maximum broadside realized gain is 6.14 dBi. Additionally, the suppression levels in both the low- and high-frequency bands exceed 15 dB. The actual test results are in good agreement with the simulation results, with only a 0.2 GHz shift toward lower frequencies, and other performance metrics show minimal differences, likely due to slight variations in the fabrication process.

As shown in the figure, the measured 10 dB impedance bandwidth is 35.9% (4.3–6.4 GHz), which is better than the simulation results. The depth of the low-frequency null is not as deep as the simulation results, but it remains below 15 dB. The high-frequency null performance is better than that of the simulation.

### 4.2. Radiation Patterns

Simulated radiation patterns were compared with CMA analysis results, showing general consistency in current directions and far-field effects. A minor discrepancy appears at 6.8 GHz, where the null effect is less pronounced than predicted likely due to the SMA connector though this has little impact on the null function, consistent with expectations.

Figure 8 presents simulated current distributions and far-field gain diagrams at key frequencies (4.2 GHz, 4.8 GHz, 6.2 GHz, 6.8 GHz), with results generally aligning with CMA analysis.

Figure 9 displays the measured and simulated radiation patterns of the filtering antenna at 4.8 GHz (a, b), 5.5 GHz (c, d), and 6.2 GHz (e, f), where (a, c, e) correspond to the E-plane and (b, d, f) correspond to the H-plane. It can be seen from the figures that the actual testing effect of the antenna is close to the simulation effect.

### 4.3. Comparisons with Other Designs

As shown in Table 2, compared with the designs in other papers, the filtering antenna proposed in this paper has the following advantages: In terms of area, it is the same as that in [6], but the relative bandwidth reaches 35.9%, which is much higher than 3.5% in [6]. Compared with other papers, the antenna in this paper has a smaller area. It can still achieve a wide operating bandwidth of 4.3–6.4 GHz ($S_{11}<-10$ dB) in a compact size, with excellent relative bandwidth performance. Moreover, it has only a single-layer structure, ensuring structural simplicity while achieving good performance.

## 5. Conclusions

In this study, a compact filtering antenna featuring two radiation nulls has been designed and fabricated, achieving good broadband performance in the Sub-6 GHz range. Characteristic mode analysis (CMA) was utilized for the design. The inherent properties of the filtering antenna modes enable the creation of a compact filtering antenna without the need for additional structures. The simulation results are closely aligned with the measured outcomes. Compared with previously reported filtering antennas, the proposed design exhibits effective filtering performance within a compact size.

## Figures and Tables

**Figure 1 sensors-25-04747-f001:**
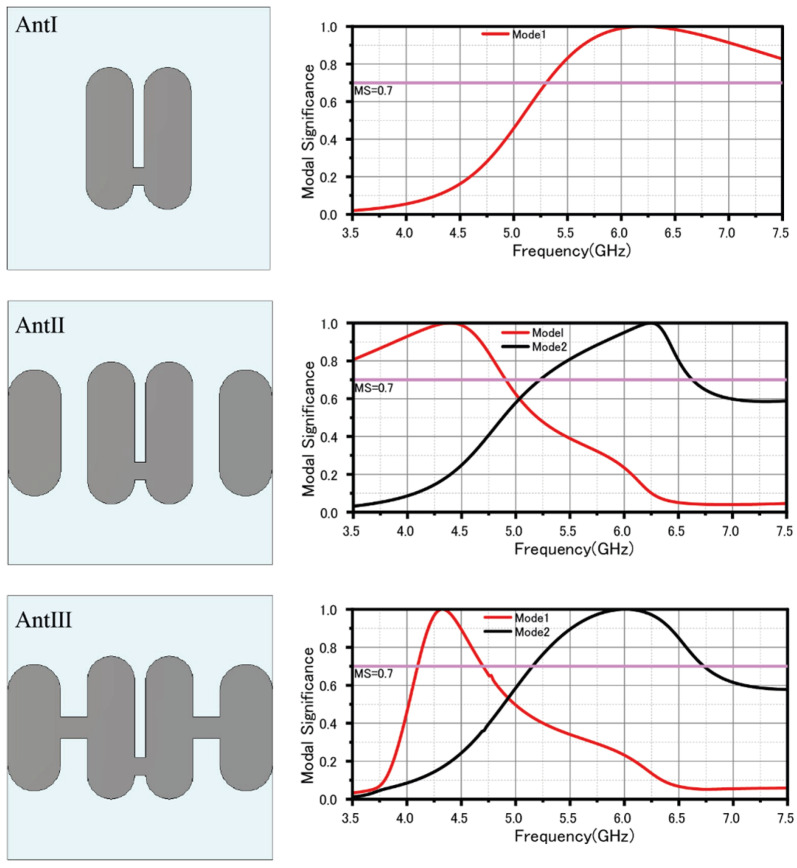
The evolutionary process of structure I, II, and III.

**Figure 2 sensors-25-04747-f002:**
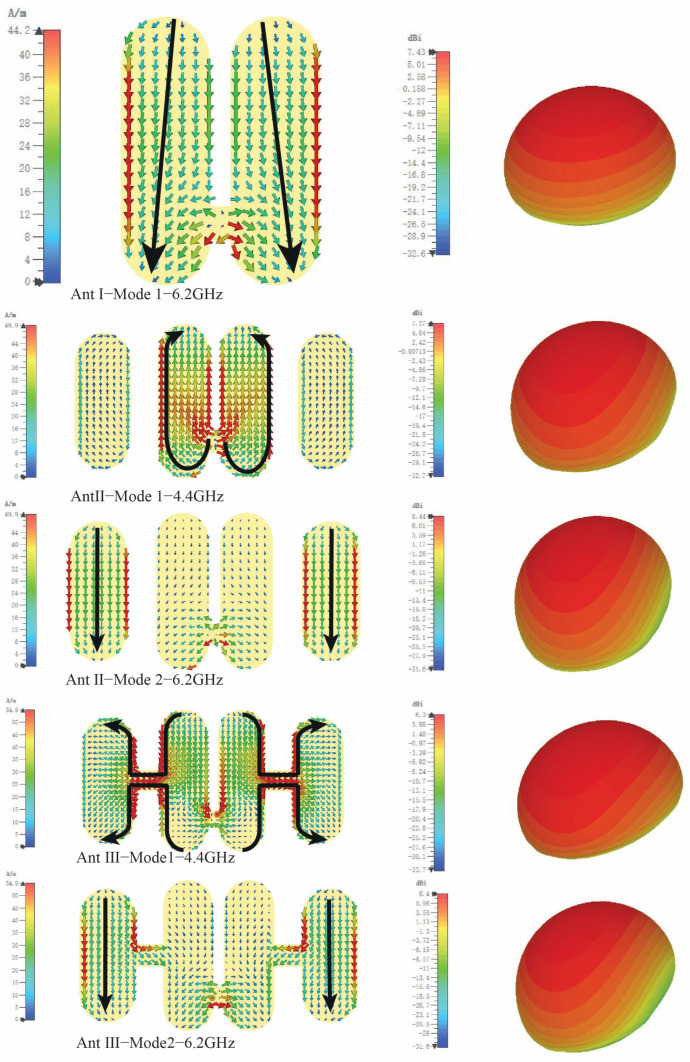
Structures I, II, and III current distribution and direction diagram.

**Figure 3 sensors-25-04747-f003:**
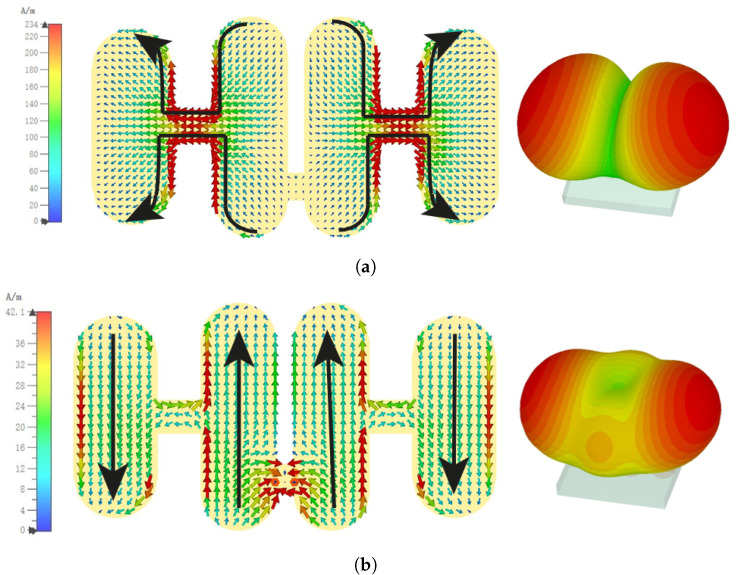
Current and radiation patterns for (**a**) Mode 1—4.2 GHz and (**b**) Mode 2—6.8 GHz.

**Figure 4 sensors-25-04747-f004:**
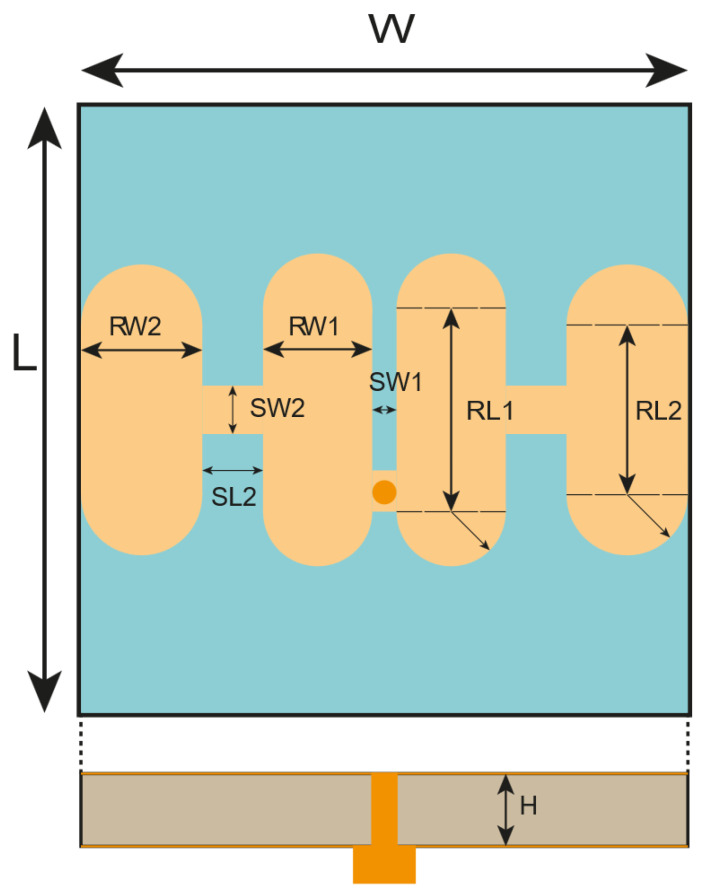
The configuration of the proposed filtering antenna.

**Figure 5 sensors-25-04747-f005:**
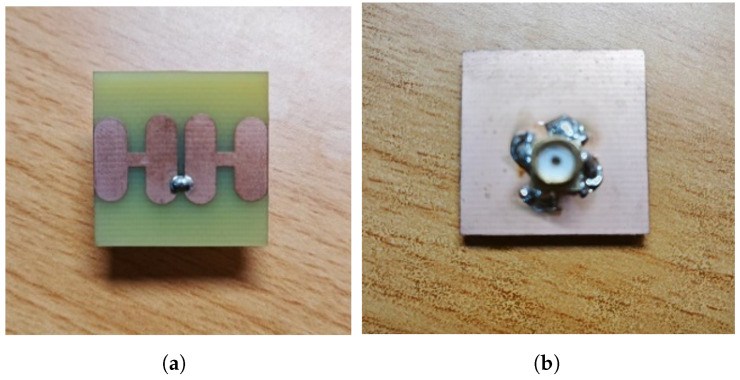
Photography of the realized antenna. (**a**) Patch side; (**b**) Feeding side.

**Figure 6 sensors-25-04747-f006:**
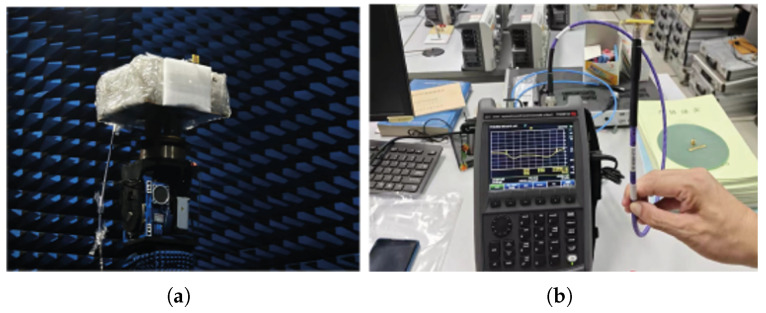
Photography of antenna actual testing environments. (**a**) Antenna far-field gain testing environment; (**b**) Return loss testing environment.

**Figure 7 sensors-25-04747-f007:**
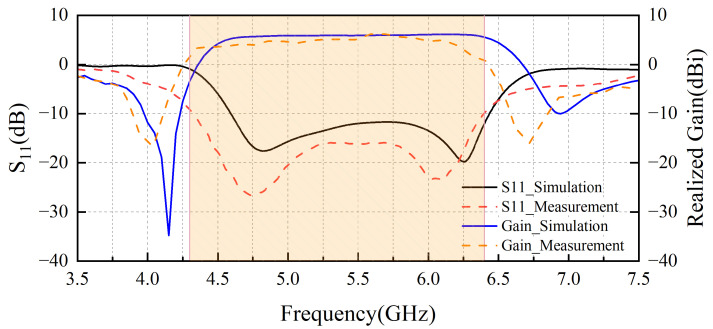
Antenna simulation and actual results.

**Figure 8 sensors-25-04747-f008:**
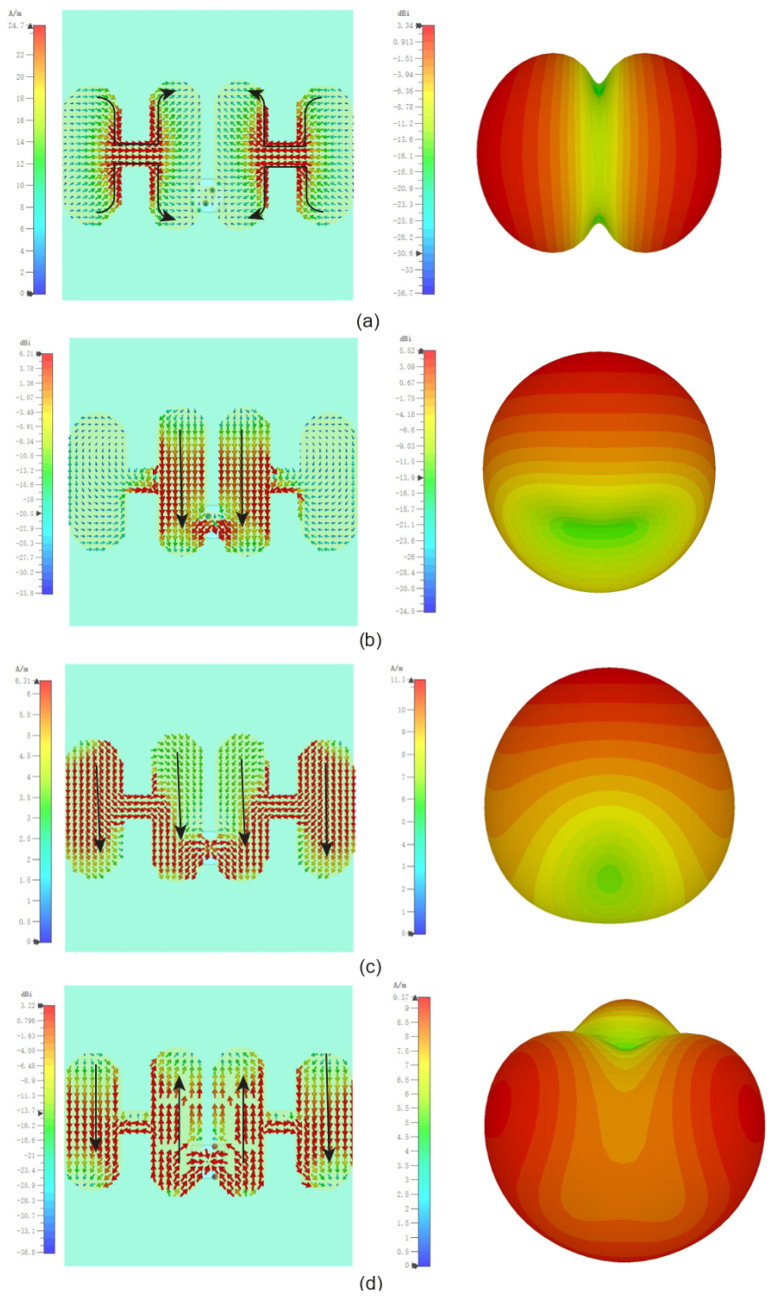
Current distribution and gain plots of the antenna at 4.2 GHz (**a**), 4.8 GHz (**b**), 6.2 GHz (**c**), and 6.8 GHz (**d**).

**Figure 9 sensors-25-04747-f009:**
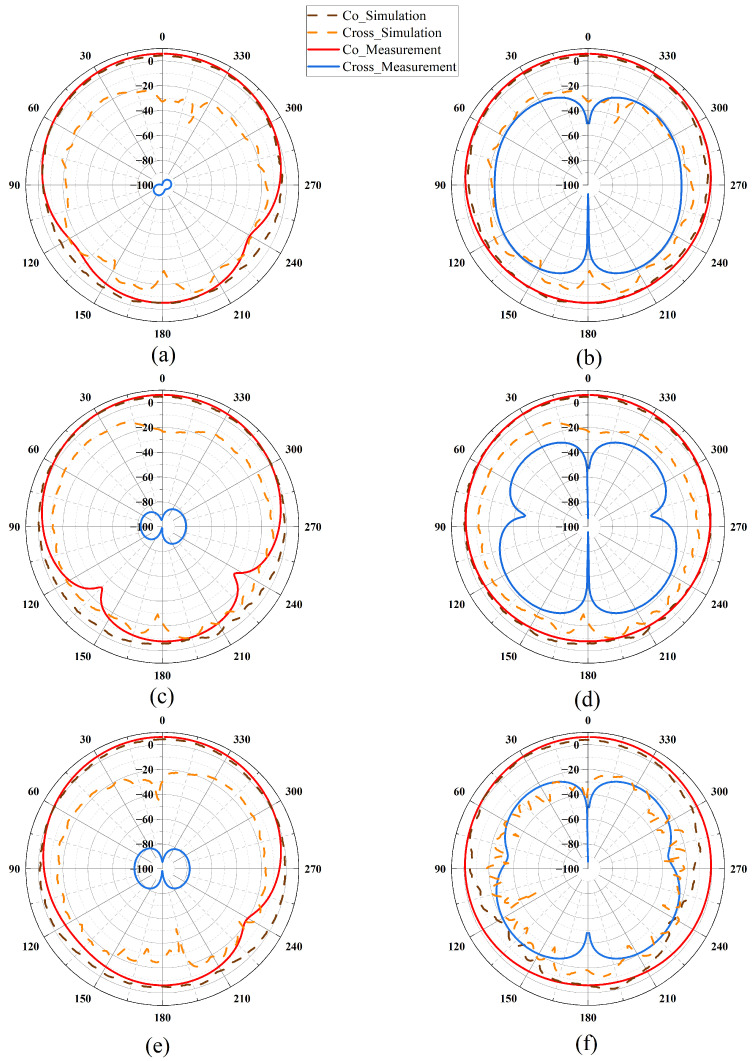
The measured and simulated radiation patterns of the filtering antenna at 4.8 GHz (**a**,**b**), 5.5 GHz (**c**,**d**), 6.2 GHz (**e**,**f**), E-plane (**a**,**c**,**e**), H-plane (**b**,**d**,**f**).

**Table 1 sensors-25-04747-t001:** The detail dimension.

Parameter	Length (mm)
RW1	9
RW2	10
RL1	8.4
RL2	7
SW1	1
SW2	2
SL1	2.5
SL2	1.7
H	3
L	25
W	25

**Table 2 sensors-25-04747-t002:** Performance comparison between proposed and reported works.

Ref.	Area ($\lambda_{0}^{2}$)	$S_{11}<-10$ dB	Gain (dBi)	Layers	Relative Bandwidth (%)
[5]	0.82×0.82	6.87–8.1	8.00	2	17.6
[6]	0.53×0.53	6.25–6.47	12.10	1	3.5
[8]	1.25×1.25	5.22–6.30	7.94	2	18.9
[7]	1.63×1.06	4.35–5.45	8.80	2	22.4
[9]	1.35×1.35	4.6–7.0	7.60	2	41.4
[11]	1.00×1.18	3.28–3.79	11.14	5	14.4
[12]	1.22×1.22	2.20–2.69	9.50	1	20.1
**Pro.**	0.45×0.45	4.3–6.4	6.14	1	35.9

## Data Availability

The original contributions presented in this study are included in the article. Further inquiries can be directed to the corresponding author.

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
