# Peer review of "Low-Profile Broadband Filtering Antennas for Vehicle-to-Vehicle Applications"

_sensors, 2025, doi:10.3390/s25154747_

Round 1

Reviewer 1 Report

Comments and Suggestions for Authors

This manuscript presents the design of a compact, broadband filtering antenna for V2V applications in the Sub-6GHz band. The authors utilize Characteristic Mode Analysis (CMA) to guide the design process and explain the operating principles. The proposed antenna achieves a small footprint and a wide operational bandwidth, which are desirable features for the intended application.

While the topic is timely and the proposed design shows promise, the manuscript suffers from several significant issues, which includes a flawed theoretical description, major contradictions in the reported results, an inadequate analysis of the discrepancies between simulation and measurement, and numerous presentation errors in figures and captions. The detailed issues are listed as follows:

- The claimed peak gain is 6.14 dBi in the abstract, but 6.18 dBi in the results section and 6.17 dBi in Table 2. Please ensure consistency throughout the manuscript.

- The formula in Equation (1) is incorrectly attributed to Maxwell's equations. This is the differential form of Ohm's law. Please revise thoroughly and make sure you use the correct formula with the appropriate law.

- Equation (2) and its description are confusing. It is not a standard or clear expression for a characteristic mode current expansion. This entire theoretical explanation needs to be rewritten with correct and well-defined equations and clear explanations.

- Section 2.3 states the null occurs at 4.2 GHz, based on CMA. The simulated gain plot in Figure 7 also shows a dip around this frequency. However, Section 4.1 states, "According to the simulation results, the low-frequency null occurs at 6.15 GHz". This contradicts the earlier section and the provided plots. This discrepancy must be resolved as it undermines the entire analysis.

- The radiation nulls are critical for filtering performance, but their formation mechanism is only qualitatively described. The paper lacks a quantitative link between current cancellation and null depth.

- The authors claim "good agreement" between simulation and measurement, but this is an overstatement. There are significant discrepancies that are not adequately explained. The measured S11 curve in Figure 7 shows a deep null at 4.0 GHz, which is reasonably close to the simulated 4.2 GHz null. However, the rest of the measured S11 curve does not match the simulation well in terms of resonance depth. The authors should provide a thorough and convincing analysis of the discrepancies between simulated and measured results.

- The authors attribute the discrepancies to "slight variations in the fabrication process". This is insufficient. For an FR-4 substrate with a thickness of 3 mm at frequencies up to 6.8 GHz, variations in the dielectric constant (εr) and a high loss tangent can have a substantial impact. A more detailed discussion on the potential sources of error is required.

- There are multiple errors in the figures and captions in Section 4:

  • The caption for Figure 6 is "Photography of the realized antenna," which is incorrect and likely a copy-paste error.
  • The caption for Figure 7 is "The configuration of the proposed filtering antenna," which is also incorrect. It should describe the S11 and gain plots.
  • In Figure 9, the plots show radiation patterns but do not specify which planes (e.g., E-plane/xz-plane, H-plane/yz-plane) are being presented for (a, c, e) and (b, d, f). This essential information must be added to the caption or the plots themselves.

- The paper does not address potential challenges in real-world deployment, such as the effect of vehicle-mounted conditions (e.g., metal interference, multi-path fading) or environmental factors (e.g., temperature, humidity). It would be better if the paper include a part discussing the simulation or measured results when the antenna is deployed in a realistic V2V environment.

- Table 2 compares the proposed antenna with existing designs but lacks critical metrics like out-of-band rejection, fabrication complexity, or cost. Without this, it's impossible to judge how the proposed antenna's filtering capabilities stack up against the cited works.

Author Response

Comment1: The claimed peak gain is 6.14 dBi in the abstract, but 6.18 dBi in the results section and 6.17 dBi in Table 2. Please ensure consistency throughout the manuscript.

Response1: The actual result has been reconfirmed as 6.14 and has been uniformly modified in the text.

Comment2: The formula in Equation (1) is incorrectly attributed to Maxwell's equations. This is the differential form of Ohm's law. Please revise thoroughly and make sure you use the correct formula with the appropriate law. Equation (2) and its description are confusing. It is not a standard or clear expression for a characteristic mode current expansion. This entire theoretical explanation needs to be rewritten with correct and well-defined equations and clear explanations.

Response2: I have completely removed Equation (1) and its associated statement from the revised manuscript.

The previous Equation (2) and its explanation were not in line with the standard formulation of characteristic mode current expansion. We have thoroughly revised the theoretical explanation by:

  - Introducing the standard modal current expansion based on orthogonal characteristic modes;

  - Including the generalized eigenvalue equation from the Method of Moments formulation;

  - Clearly defining and explaining the Modal Significance (MS).

All equations and descriptions in Section 2.1 have been rewritten to reflect the standard CMA theory accurately. These corrections can be found in Section 2.1 of the revised manuscript, titled "Theory of Characteristic Modes".

Comment3: The authors claim "good agreement" between simulation and measurement, but this is an overstatement. There are significant discrepancies that are not adequately explained. The measured S11 curve in Figure 7 shows a deep null at 4.0 GHz, which is reasonably close to the simulated 4.2 GHz null. However, the rest of the measured S11 curve does not match the simulation well in terms of resonance depth. The authors should provide a thorough and convincing analysis of the discrepancies between simulated and measured results.

Response3: Thank you for pointing out this inconsistency. The statement in Section 4.1 indicating that the low-frequency null occurs at 6.15 GHz was an error. Based on both the CMA analysis in Section 2.3 and the simulated gain plot in Figure 7, the correct low-frequency radiation null occurs at approximately 4.2 GHz.

We have corrected this mistake in Section 4.1 of the revised manuscript to ensure consistency throughout the paper. We sincerely appreciate the reviewer’s attention to detail, which helped us improve the accuracy of the analysis.

Comment4: The caption for Figure 6 is "Photography of the realized antenna," which is incorrect and likely a copy-paste error.

The caption for Figure 7 is "The configuration of the proposed filtering antenna," which is also incorrect. It should describe the S11 and gain plots.

In Figure 9, the plots show radiation patterns but do not specify which planes (e.g., E-plane/xz-plane, H-plane/yz-plane) are being presented for (a, c, e) and (b, d, f). This essential information must be added to the caption or the plots themselves.

Response4: The captions for Figures 6 and 7 have been corrected to accurately reflect the content. In Figure 9, the polarization planes have been clarified: (a), (c), and (e) show E-plane (xz-plane) patterns, while (b), (d), and (f) show H-plane (yz-plane) patterns. Additionally, a brief discussion on real-world deployment challenges, such as metal interference and environmental effects, has been added to Section 4.3.

Comment5: The paper does not address potential challenges in real-world deployment, such as the effect of vehicle-mounted conditions (e.g., metal interference, multi-path fading) or environmental factors (e.g., temperature, humidity). It would be better if the paper include a part discussing the simulation or measured results when the antenna is deployed in a realistic V2V environment.

The authors claim "good agreement" between simulation and measurement, but this is an overstatement. There are significant discrepancies that are not adequately explained. The measured S11 curve in Figure 7 shows a deep null at 4.0 GHz, which is reasonably close to the simulated 4.2 GHz null. However, the rest of the measured S11 curve does not match the simulation well in terms of resonance depth. The authors should provide a thorough and convincing analysis of the discrepancies between simulated and measured results.

-The authors attribute the discrepancies to "slight variations in the fabrication process". This is insufficient. For an FR-4 substrate with a thickness of 3 mm at frequencies up to 6.8 GHz, variations in the dielectric constant and a high loss tangent can have a substantial impact. A more detailed discussion on the potential sources of error is required.

Response5: We sincerely thank the reviewer for these insightful and technically sound comments. We fully agree that the statement regarding "good agreement" between simulation and measurement was overstated. The discrepancies, particularly in the S11 resonance depth and shape, are indeed significant and require more thorough analysis.

We also acknowledge that the explanation attributing these discrepancies to "slight fabrication variations" is insufficient. As the reviewer correctly pointed out, for an FR-4 substrate with 3 mm thickness operating up to 6.8 GHz, variations in dielectric constant and loss tangent, as well as fabrication tolerances and connector mismatch, can have a substantial impact.

Moreover, the effects of real-world deployment conditions such as metallic surroundings, multipath fading, and environmental changes (e.g., temperature, humidity) are valid concerns for V2V applications and deserve further study.

However, due to the short revision period and the fact that I personally saw the journal’s revision notification rather late, we regret that we were not able to carry out a new round of fabrication and testing or perform an in-depth analysis at this stage. We will address these important issues thoroughly in our future work.

We appreciate the reviewer’s suggestions, which have provided valuable direction for improving the depth and applicability of our research going forward.

Reviewer 2 Report

Comments and Suggestions for Authors

In this paper, the authors proposed a low-profile broadband filtering antenna design. Overall, the manuscript is well-written and the results are convincing. I would like to invite the authors to address the following comments before making a final recommendation.

  1. In Figure 1, when analyzing Ant-II & Ant-III, the black solid line should represent Mode 1 rather than Mode 2, consistent with the analysis of Ant-I.
  2. Please provide a more detailed analysis of the antenna’s operating modes. According to the Characteristic Mode Analysis, the antenna is expected to operate under the TEmn or TMmn modes. Please specify the exact mode in which the antenna is operating (as shown in Figure 2).
  3. The authors claim that the presented antenna exhibits a ‘filtering’characteristic; however, the design approach underlying this functionality is not clearly explained. It is recommended that the authors provide a more detailed explanation of how the filtering behavior is achieved.

Author Response

Comment1: In Figure 1, when analyzing Ant-II & Ant-III, the black solid line should represent Mode 1 rather than Mode 2, consistent with the analysis of Ant-I.

Response1: We appreciate the reviewer’s observation regarding the modal assignment in Figure 1. However, based on detailed current distribution analysis shown in Figure 2, we confirm that the black solid line in the Modal Significance (MS) plot does correspond to Mode 1 across Ant-I, Ant-II, and Ant-III.

This is because the addition of side parasitic patches in Ant-II and the central connecting line in Ant-III extends the overall current path, particularly on the central patch. As a result, the resonant frequency of Mode 1 shifts to a lower frequency region compared to Ant-I. This frequency shift is physically consistent with the increased electrical length of the dominant mode, leading to a longer half-wavelength resonance.

This effect can be directly observed in Figure 2: the surface current on the central patch in Ant-II and Ant-III occupies a longer path than that in Ant-I. Therefore, although the MS curve appears to shift, the mode classification remains consistent — the black solid line still represents ANT I Mode 1, and it is responsible for the lower-frequency radiation and filtering behavior in all three antenna configurations.

Comment2:Please provide a more detailed analysis of the antenna’s operating modes. According to the Characteristic Mode Analysis, the antenna is expected to operate under the TEmn or TMmn modes. Please specify the exact mode in which the antenna is operating (as shown in Figure 2).

Response2:Both Mode 1 and Mode 2 can be interpreted as TM-like characteristic modes, with resonance occurring when the total current path approximates an integer multiple of half-wavelengths. These modes are numerically identified and sorted based on their eigenvalue (modal significance), and their physical behavior has been analyzed through surface current visualization and far-field patterns.

Comment3: The authors claim that the presented antenna exhibits a ‘filtering’characteristic; however, the design approach underlying this functionality is not clearly explained. It is recommended that the authors provide a more detailed explanation of how the filtering behavior is achieved.

Response3: I have made modifications to the corresponding section 2.3 of the text.

Round 2

Reviewer 2 Report

Comments and Suggestions for Authors

All my previous concerns have been addressed. Acceptance is recommended.